# Iron Isotope Constraints on the Mineralization Process of Shazi Sc-Rich Laterite Deposit in Qinglong County, China

**Jun Sun [1], Yunlong Liu [2] and Xiqiang Liu [3,4,*]**

1  The Department of Resource and Environmental Engineering, Guizhou Institute of Technology, Guiyang 550003, China; sunjun531@163.com
2  The School of Management Science, Guizhou University of Finance and Economics, Guiyang 550025, China; lyl007tt@163.com
3  Key Laboratory of High-Temperature and High-Pressure Study of The Earth's Interior, Institute of Geochemistry, Chinese Academy of Sciences, Guiyang 550081, China
4  University of Chinese Academy of Sciences, Beijing 100049, China
*  Correspondence: liuxiqiang@mail.gyig.ac.cn; Tel.: +86-150-2997-4006

**Abstract:** The Shazi deposit is a newly discovered, potential large-scale scandium deposit located in Qinglong, southwestern Guizhou Province, China. The iron isotopic composition of magnetite in fresh basalt, weathered basalt, and mineralized laterite was investigated. The Fe content of fresh basalt and of weathered basalt vary from 15.41 wt.% to 15.51 wt.% and from 14.60 wt.% to 15.12 wt.%, respectively, while the $\delta^{56}$Fe of magnetite varies from 0.23‰ to 0.29‰ and from 0.02‰ to 0.07‰. Laterite has the highest Fe content, in the range of 23.53%~28.95%, but $\delta^{56}$Fe is similar to weathered basalt, and the range of variation is −0.09‰–0.03‰. The change in iron isotope composition in weathered basalt and laterite is related to the hydrolysis of clinopyroxene. Combined with the existing research results, the genesis of scandium deposit is considered to be related to in situ hydrolysis in deep and surface weathering leaching of Emeishan basalt.

**Keywords:** laterite; scandium deposit; iron isotope; Emeishan basalt



## 1. Introduction

Scandium (Sc) is known as a high-tech metal. It is mainly used in the preparation of Sc-Al alloy series, Sc-Na lamp, solid oxide fuel cell, special steel, nonferrous alloy, high-performance ceramics, etc. and as a catalyst to the increasing demand for Sc in many emerging strategic industries; therefore, it is listed as a critical metal by China [1] and the EU [2]. Based on the enrichment processes, the primary Sc deposits can be divided into magmatic ore deposits, hydrothermal ore deposits, and supergene ore deposits [3]. The grade of Sc in these primary deposits is in the range of 0.005–0.04 wt.% [3]. Only 12 terrestrial minerals are known to contain Sc as an essential component, and thortveitite [$Sc_2Si_2O_7$] is the most important among these minerals [4,5]. Significant proportions of Sc can be hosted in ferromagnesian silicate minerals (including clinopyroxene and garnet) and in a number of high-field-strength-element (HFSE) minerals such as wolframite baddeleyite, rutile, etc. [3,4,6–8]. At the beginning of the 20th century, Sc was mined from the thortveitite-bearing pegmatite in Evje-Iveland, Norway [4]. Later, Sc was mainly recovered as a by-product from residues, tailings, and waster liquors in the production of rare earth element (REE), tungsten, and titanium [9]. Recently, about 90% of the global Sc production is from the Bayan Obo REE mine, the largest REE deposit in the world [4].

Recently, the Sc-rich laterite was discovered in Cuba and the Dominican Republic [10], Australia [11], New Caledonia [12], NE Argentina [13], and China [14]—which may be a potential Sc resource in the future. The laterite-type Sc deposit has attracted much attention. The resource potential of this type of deposit is significant, and scandium can be mined as an independent deposit; further, the technology of extracting scandium from laterite is feasible

and economical [11]. Based on the study of the world-class lateritic scandium deposit in Australia, accounting for the close symbiotic relationship of iron and scandium, Chassé et al. (2017) [11] considered that scandium is mostly adsorbed on goethite, and few substituted into the hematite in the laterite. By a combination of sequential extraction, synchrotron-based microfocused X-ray fluorescence and X-ray absorption fine structure techniques, Qin et al. (2020; 2021) confirmed that Sc can be structurally incorporated into goethite apart from the adsorption on the goethite surface [15,16]. Iron (oxyhydr)oxides, such as goethite, hematite, and magnetite, form an important component of laterite. Previous studies have shown that the change of iron isotope can effectively indicate the geochemical behavior of iron in the process of soil formation and trace the redox environment of soil evolution [17–21]. Therefore, through the study of iron isotope and iron content in laterite, we can examine the geochemical behavior of iron in the formation and evolution of laterite and then determine the enrichment and mineralization mechanism of scandium.

The recently discovered Qinglong Shazi laterite-type deposit in southwestern Guizhou, China, is a potential large-scale Sc deposit that may be related to the laterization of Emeishan basalt [10]. Previous studies mainly focused on the geological background, deposit characteristics, and ore-forming material sources of the deposit [11], but the enrichment process of scandium is not well constrained. Based on the iron isotope study of magnetite in fresh basalt, weathered basalt, and laterite in the area, combined with the whole-rock iron content data, this paper aims to find out the changes of iron isotope and iron content in the basalt laterization process and to investigate the enrichment process of scandium.

## 2. Geological Background

The Qinglong Shazi deposit in Guizhou is located outside the Emeishan basalt transition zone. The exposed strata of the mining area are the Middle Permian Maokou Formation, the Upper Permian Emeishan basalt formation, the Longtan Formation, and Quaternary sediments (Figure 1) [22].

Quaternary alluvial proluvial deposits are mainly distributed in low-lying gullies and karst valleys. The lithology is yellow, variegated gravel and sand, with a loose structure and angular unconformity contact with underlying strata. Quaternary residual deluvial deposits are mainly distributed in hills or microdepressions in the slope zone on the relatively gentle karst denudation surface in the mining area. The lithology is mainly red clay and sub clay, which often contain breccia. The breccia is mainly composed of basalt, siliceous limestone, siliceous rocks, and tuff, with different sizes of gravel. The lower part of the eluvial deluvial deposit is often embedded in microkarst forms such as rock bud and karst ditch, which are the main occurrence horizon of scandium deposit and have angular unconformity contact with the underlying Maokou Formation. The Upper Permian Longtan Formation is distributed in the west and northwest of the study area, with only local distribution in the area. The lithology is grayish-green, gray, and brownish-yellow, thin-to-medium-thick mudstone with sandstone. The lower part is a gray, medium-thick aluminous mudstone and siliceous rock. The middle and lower parts are intercalated with 2–3 layers of coal and sandy mudstone. It has pseudoconformity contact with the underlying Upper Permian Emeishan basalt formation. It is composed of basaltic lava, basaltic lava breccia, pyroclastic rock, tholeiitic basalt, and tuff. Emeishan basalt is widely distributed in the study area. The formation and underlying strata are in unconformity contact. The Middle Permian Maokou Formation is distributed among most parts of the study area. The lithology is bright crystal bioclastic limestone and limestone, and these limestones are gray and dark gray in color.

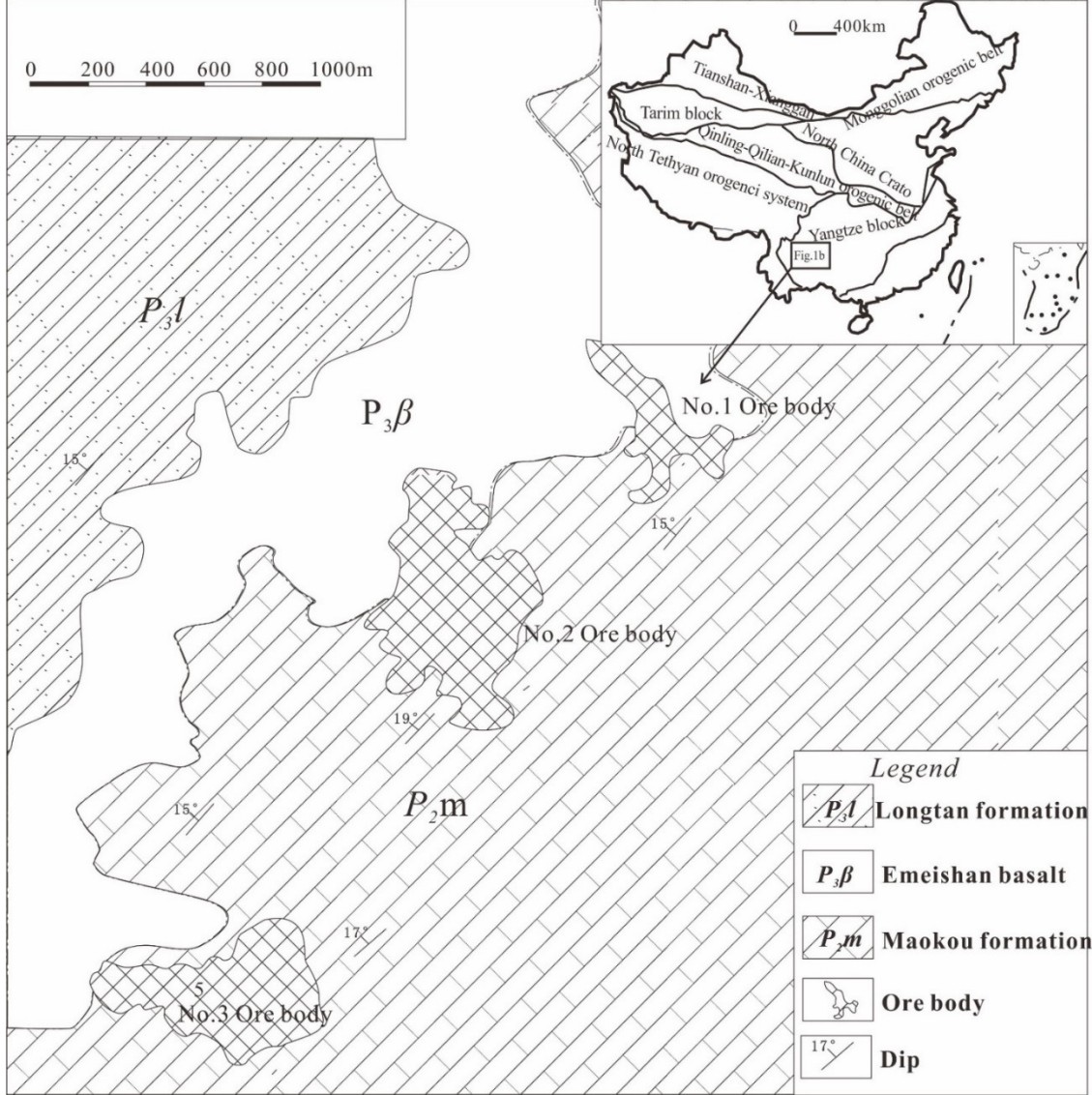

**Figure 1.** Geological sketch of the Shazi laterite-type Sc deposit.

In the middle of the last century, the Shazi deposit was mined for $TiO_2$, and the deposit was mined out at the end of the last century. In 2007, the research on the deposit found that the content of scandium in laterite was very high, reaching the cutoff grade (>0.005 wt.%); thus, it was defined as a laterite-type Sc deposit [22], but it was not mined at present. The Sc deposit occurs in the Quaternary eluvial slope red soil above the karst unconformity of Maokou limestone in the Middle Permian. Spatially, Sc ore bodies are arranged in the direction of NE–SE (Figure 1), and in the following order: No.1 ore body, No.2 ore body, and No. 3 ore body [22].

(1) No.1 ore body: The ore body is irregular in the NW–SE direction on the surface, and the profile is stratoid. The surface distribution area is about 71,655 m$^2$, 498–665 m long, and 21–60 m wide. It is the smallest ore body in the deposit with a thickness of 4.4–22.5 m. The average grade of $Sc_2O_3$ is ~0.0067 wt.%. The resource of $Sc_2O_3$ is 148 t.

(2) No.2 ore body: The ore body has irregular lenticular distribution in the direction of NNW–SE on the surface, and the profile is stratoid. The surface distribution area of the ore body is about 297,982 m$^2$, which is the largest ore body in the deposit, with a length of 580–955 m, a width of 93–590 m, and a thickness of 2.7–42.0 m. The average grade of $Sc_2O_3$ is 0.0073 wt.%, and the resource of $Sc_2O_3$ is 948 t.

(3) No.3 ore body: The ore body is irregularly distributed in approximately the east–west direction on the surface, and the profile is stratoid. The surface distribution area of the ore body is about 204,135 m$^2$, 320–789 m long, 155–465 m wide, 3.5–24.8 m thick, and the average grade of $Sc_2O_3$ is 0.0083 wt.%. The resource of $Sc_2O_3$ is 650 t.

## 3. Samples and Analytical Methods

### 3.1. Samples

Eight samples were collected from drill holes in the No.1 ore body (N25°46′40.72″; E105°9′0.01″): two samples from fresh pillow basalt, two samples weathered basalt, and four from laterite. The samples were crushed to a particle size of 60–100 mesh. Magnetite was separated using a magnetic separator, then carefully handpicked under a binocular microscope. The purity of the separated mineral fraction was greater than 95%. The individual grains were cleaned ultrasonically three times in ultrapure water for 10 min.

### 3.2. Analytical Methods

Fe isotope analyses were performed at the State Key Laboratory of Ore Deposit Geochemistry, Institute of Geochemistry, Chinese Academy of Sciences. The detailed procedures for sample dissolution, column chemistry, and instrumental analysis were introduced by Zhu et al. (2008) [23]. A brief description is given below.

#### 3.2.1. Acid Digestion of Magnetite Grains

A total of 3 milligrams of the hand-picked magnetite fraction was digested with 3 mol/L concentrated hydrochloric acid at 120 °C for 3 h.

#### 3.2.2. Iron Extraction and Purification

AG MP-1 Anion exchange resin was used to separate Fe from the solution. Prior to usage, the AG MP-1 Anion exchange resin was soaked in Milli-Q ultrapure water and packed into a column by the wet method. The column was alternately washed several times with 0.5 mol/L $HNO_3$ and Milli-Q ultrapure water, then equilibrated with 7 mol/L HCl + 0.001% $H_2O_2$. A quantity of 0.2 mL of sample solution was poured into the column, where the matrix elements were removed with 25 mL of 7 mol/L HCl + 0.001% $H_2O_2$, then eluted with 22 mL of 2 mol/L HCl + 0.001% $H_2O_2$ reagent to extract Fe. The eluent with Fe was evaporated and dissolved in a diluted nitric acid medium for mass spectrometry.

#### 3.2.3. Iron Isotope Measurement

The iron isotopic composition was determined by Nu instruments high-resolution multi-collector–inductively coupled plasma mass spectrometry (HR-MC-ICPMS). The instrument can effectively remove the interference of $^{40}Ar^{14}N$, $^{40}Ar^{16}O$, and other polyatomic ion groups on $^{54}Fe$ and $^{56}Fe$.

In the process of analysis, the "standard–sample–standard" cross method was used to correct the mass fractionation of the instrument, and the relative deviation of concentration between the standard sample and the sample solution was controlled within 10%. After sample and standard signal acquisition, the pump tubing and nebulizer were washed out with 10% and 0.1% nitric acid for 3 min and 2 min, respectively. The signals of Fe isotopes were simultaneously received by three Faraday cups in static mode. The data were collected automatically by software, and each group of data was tested for 20 s before collection.

For the iron isotope, IRMM-014—provided by the Institute of reference materials and measurement of the European Commission—is often used as the isotope standard. The Fe isotopic abundance of IRMM-014 is as follows: 5.845 ± 0.023% ($^{54}Fe$), 91.754 ± 0.024% ($^{56}Fe$), 2.1192 ± 0.0065% ($^{57}Fe$), 0.2818 ± 0.0027% ($^{58}Fe$) [24]. The results of Fe isotope analysis were calculated by the thousandth deviation relative to IRMM-014 $\delta^x Fe$:

$$\delta^{56}Fe \ (‰) = [(^{56}Fe/^{54}Fe, \text{sample})/(^{56}Fe/^{54}Fe, \text{IRMM-014}) - 1] \times 10^3 \qquad (1)$$

$$\delta^{57}Fe \ (‰) = [(^{57}Fe/^{54}Fe, \text{sample})/(^{57}Fe/^{54}Fe, \text{IRMM-014}) - 1] \times 10^3 \qquad (2)$$

## 4. Results

The Fe isotopic compositions of magnetite from, pillow basalt, weathered basalt, and laterite are presented in Table 1. The $\delta^{56}$Fe of magnetite from the Maokou Formation limestone is $0.13 \pm 0.05$‰. The $\delta^{56}$Fe of the magnetite from fresh pillow basalt ranges from 0.23‰ to 0.29‰, which are the heaviest Fe-isotope signatures of the investigated samples. The magnetites from weathered basalts and laterite are characterized by a slightly lighter Fe isotopic composition, in which $\delta^{56}$Fe ranges from 0.02‰ to 0.07‰, and from $-0.09$‰ to 0.03‰, respectively.

**Table 1.** Magnetite iron isotopic analytical results of the Shazi anatase deposit.

| Rock Type | No. | $\delta^{57}$Fe | | $\delta^{56}$Fe | | TFe$_2$O$_3$ * |
|---|---|---|---|---|---|---|
| | | Ratio | 2SD | Ratio | 2SD | |
| Pillow basalt | T-002 | 0.32 | 0.06 | 0.23 | 0.07 | 15.51 |
| Pillow basalt | T-003 | 0.42 | 0.07 | 0.29 | 0.06 | 15.41 |
| Weathered basalt | 2-002 | 0.05 | 0.12 | 0.02 | 0.07 | 15.12 |
| Weathered basalt | 2-004 | 0.13 | 0.10 | 0.07 | 0.05 | 14.60 |
| Laterite | 2-011 | −0.07 | 0.09 | −0.04 | 0.05 | 23.53 |
| Laterite | 2-012 | −0.03 | 0.10 | −0.02 | 0.05 | 23.59 |
| Laterite | 3-003 | −0.10 | 0.12 | −0.09 | 0.06 | 28.95 |
| Laterite | 3-004 | 0.02 | 0.09 | 0.03 | 0.07 | 24.46 |

* data from [15].

## 5. Discussion

### 5.1. Implications of Fe Isotope

When plotted on a three-isotope diagram, all iron isotopic data define a single mass-fractionation line with a slope of ~0.70. The correlation between $\delta^{56}$Fe and $\delta^{57}$Fe is $\delta^{56}$Fe $= 0.703 \times \delta^{57}$Fe, where $R^2 = 0.986$ (Figure 2).

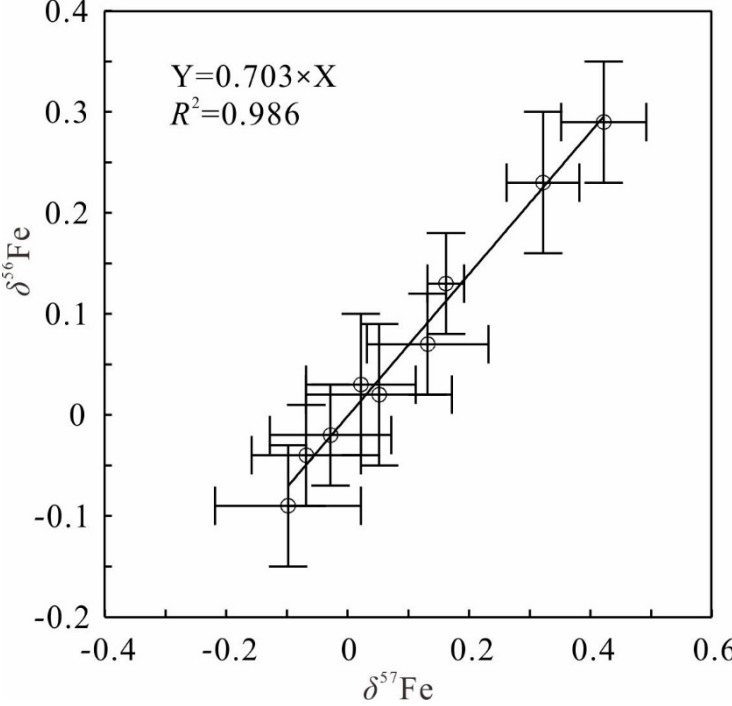

**Figure 2.** The diagram of two Fe isotopes.

According to XRD (the data are available in Supplementary Materials), fresh basalt is mainly composed of clinopyroxene and plagioclase, and iron oxides include magnetite and ilmenite (in heavy sand minerals). In addition to clinopyroxene, plagioclase, and magnetite, montmorillonite appears in weathered basalt, while laterite is mainly composed of clay minerals and iron (oxyhydr)oxides (such as goethite and hematite). Magnetite appears in heavy sand minerals and is the only mineral that exists in all three lithologies.

Chen et al. [25] analyzed the Fe isotopic composition of the whole rock, magnetite, ilmenite, olivine, and clinopyroxene from the Baima gabbro-layered intrusion, which belong to the Emeishan basalt Formation. The average value of $\delta^{56}Fe$ in the whole rock is 0.04‰. Clinopyroxene, olivine, and ilmenite have relatively light Fe isotopic compositions. The values of $\delta^{56}Fe$ are −0.04‰, −0.02‰, and −0.10‰, respectively. Magnetite has a heavy isotopic composition. The value of $\delta^{56}Fe$ in magnetite is 0.29‰. The above data show that magnetite is different from other Fe-bearing minerals in the isotopic composition, and the isotope of magnetite has changed significantly. Therefore, the variation of the iron isotope in magnetite is an effective indicator of the whole-rock iron isotope, the composition of which is essentially the result of mixing Fe isotopic compositions of various Fe-bearing mineral phases; thus, the main Fe-bearing mineral phase (such as clinopyroxene Fe isotopic composition) is the key factor to determine the whole-rock iron isotopic composition. As shown in Figure 3, The $\delta^{56}Fe$ of magnetite in fresh basalt from the Shazi Sc deposit is in the range of 0.23‰ to 0.29‰, with an average of 0.26‰, which is basically the same as that of magnetite in Baima gabbro. The magnetite in weathered basalt has a lower $\delta^{56}Fe$ value, but the content of $TFe_2O_3$ is almost the same, whereas the $TFe_2O_3$ content in laterite significantly increases, but the $\delta^{56}Fe$ of magnetite in laterite is similar to that of weathered basalt.

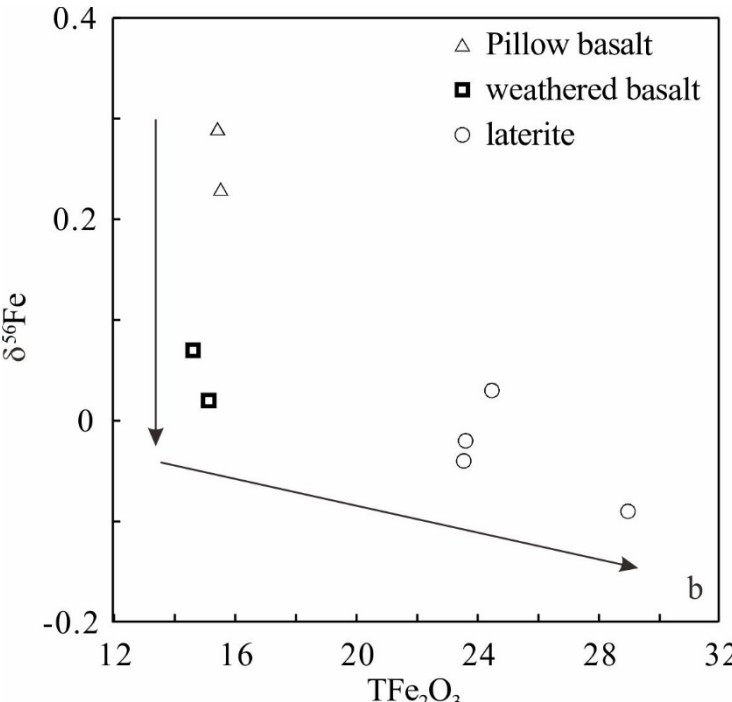

**Figure 3.** Relationship plot between the Fe isotopic composition of magnetite and the Fe content in the Qinglong Shazi Sc deposit. T stands for total Fe.

During basalt hydrolysis and weathering in laterite, Fe undergoes dissolution, oxidation (or reduction), precipitation (or migration), and other processes. In the above process, both Fe content and isotope composition may change. The main reason for these changes is the decomposition of Fe-bearing minerals, especially clinopyroxene. With the decomposition of clinopyroxene, $Fe^{2+}$ enters the aqueous solution. If $Fe^{2+}$ migrates with the solution

due to its high mobility, the content of Fe in the system should be significantly reduced. In reality, the weathered basalt and fresh basalt almost have the same Fe content, which suggests that $Fe^{2+}$ may be in situ oxidized to $Fe^{3+}$, and secondary iron oxides (such as goethite, hematite, and/or magnetite) are formed. Under equilibrium fractionation, the Fe isotope fractionation between clinopyroxene, $Fe(II)_{aq}$ in aqueous solution, and secondary magnetite should have the following expression:

$$\Delta^{56}Fe(II)_{aq\text{-}magnetite} = \delta^{56}Fe(II)aq - \delta^{56}Fe_{magnetite}. \tag{3}$$

$$\Delta^{56}Fe(II)_{aq\text{-}clinopyroxene} = \delta^{56}Fe(II)_{aq} - \delta^{56}Fe_{clinopyroxene}. \tag{4}$$

$$\delta^{56}Fe_{\,magnetite} = \delta^{56}Fe_{clinopyroxene} + \Delta^{56}Fe(II)_{aq\text{-}clinopyroxene} - \Delta^{56}Fe(II)_{aq\text{-}magnetite}. \tag{5}$$

$$\delta^{56}Fe_{magnetite} = \delta^{56}Fe_{clinopyroxene} + \Delta^{56}Fe(II)_{aq\text{-}clinopyroxen\,e} - \Delta^{56}Fe(II)a_{q\text{-}magnetite} = 0.145 \times 106/T^2 - 1.07. \tag{6}$$

The combination of the above two formulas is as follows:

The Fe isotopic fractionation equation between magnetite and hydrothermal fluid is $\Delta^{56}Fe(II)_{aq\text{-}magnetite} = -0.145 \times 106/T^2$ (T is in K) + 0.10 [26]. The values of $\Delta^{56}Fe(II)_{aq\text{-}clinopyroxene}$ vary from $-1.50‰$ to $0.50‰$ [27]. In this paper, we have taken the average value of $-1.00‰$. By substituting the value into Equation (5), the relationship between the Fe isotope of magnetite and the temperature of alteration fluid can be further obtained as follows:

According to the iron isotopic composition of magnetite in weathered basalt ($\delta^{56}Fe$ magnetite = 0.02–0.07‰), it is estimated that the temperature of alteration fluid is between 80 °C and 90 °C. Noteworthily, the magnetite in weathered basalt is a mixture of primary magnetite and secondary magnetite. The average value of $\delta^{56}Fe$ of the original magnetite is 0.26‰, and the value of $\delta^{56}Fe$ of the mixed magnetite is between 0.02‰ and 0.07‰; therefore, the $\delta^{56}Fe$ of the secondary magnetite should be less than 0.02‰. As a result, the true temperature of the alteration fluid is higher than estimated. Compared with fresh basalt, the Fe isotope of weathered basalt magnetite has changed, which is due to the decomposition of clinopyroxene into secondary magnetite under hydrothermal alteration. However, in the process of fluid alteration, the iron content of the whole rock shows no significant change, suggesting a closed system where no matter is brought in or out.

When basalt is completely hydrolyzed and weathered into laterite, and clinopyroxene is completely hydrolyzed, the secondary magnetite should inherit the iron isotopic composition of clinopyroxene under the closed condition, and the Fe isotopic composition of magnetite should be similar to that of clinopyroxene. The iron isotopic composition of magnetite in laterite varies from $-0.09‰$ to $0.03‰$, with an average value of $-0.03‰$, which is remarkably close to the iron isotopic composition of clinopyroxene in fresh basalt ($\delta^{56}Fe = -0.04‰$). Compared with fresh basalt, the content of Fe in laterite significantly increased. As mentioned before, during pyroxene hydrolysis, the $Fe^{2+}$ was oxidized to $Fe^{3+}$ and then precipitated in situ. The increase in iron content is likely caused by the migration of other soluble substances such as Si, Na, and Ca (see the data are available in Supplementary Materials) [28].

In summary, according to the variations in Fe isotope composition and content, the basalt transforms into laterite in the Shazi mining area, and it should undergo two stages: (1) the basalt reacts with the fluid (when the temperature is higher than 90 °C) under the condition of oxidation (dominated by $Fe^{3+}$) within a closed system (no mass exchange with the outside), resulting in full hydrolyzing of clinopyroxene; (2) the leaching stage. During this stage, minerals such as clinopyroxene have been completely hydrolyzed to form secondary iron oxides. Soluble substances (such as Si, Na, and Ca) migrate out of the system. Iron and ore-forming elements such as titanium are preserved and relatively enriched.

*5.2. Enrichment and Mineralization of Scandium*

In recent years, the enrichment of Sc has been found in some weathered laterite. Aiglsperger et al. (2016) [10] found that there was a significant positive correlation between Sc and Fe in laterite; they speculated that Sc mainly existed on the surface of iron oxide in the form of adsorption. Chassé et al. (2017) [11] and Qin et al. (2020; 2021) considered that Sc is mainly adsorbed on iron oxides, especially on goethite surfaces. Chassé et al. (2019) [29] further pointed out that since Sc is similar to Fe and Mg in crystals' chemical properties, it often enters pyroxene minerals as an isomorphism. When pyroxene is hydrolyzed to form clay minerals and iron oxides, Sc is often adsorbed on the surface of iron oxides; therefore, Chassé et al. (2017) [11] proposed that the high concentration of Sc in laterite is restricted by the following factors: (1) the high Sc content in the parent rock; (2) in the process of long-term weathering, Sc that is released by host minerals is absorbed by iron oxides and precipitates in situ without migration.

According to the iron isotopic composition of magnetite and the variation of iron content in the whole rock obtained in this study, combined with the existing research results, the mineralization process of the Qinglong sand–laterite-type independent scandium deposit can be summarized as follows:

(1) In situ hydrolysis process: The parent rock of Qinglong sandy laterite is Emeishan basalt, and its genesis is related to the mantle plume. The limestone of the Maokou Formation in the Middle Permian is affected by the active crustal uplift of the mantle plume. The top of the limestone is exposed to weathering and denudation, resulting in karstification and forming a variety of karst landforms. Due to the nearshore tidal flat, there is water in the karst depression. Subsequently, with the strong eruption of Emeishan basalt, the karst depression is filled with basalt [30]. Affected by the tectonic movement from the Late Permian to the Paleogene, the Qinglong area experienced a stage of rapid burial and then slow denudation, with a burial history of more than 100 Ma. The burial depth of the Qinglong antimony deposit near Qinglong Shazi is about 2 km, and the temperature is about 100 °C [31]. Basalt in the karst depression of the Qinglong Shazi area experienced burial for more than 100 Ma after the eruption. In this process, clinopyroxene was fully hydrolyzed by the hydrothermal solution, and Sc in clinopyroxene was released and then absorbed by secondary iron oxide. In the process of in situ hydrolysis, the system is in a closed state, where no matter moves in or out (the data are available in Supplementary Materials). The enrichment of Sc is not obvious, and only changes from one occurrence state to another. For example, the content of Sc in fresh basalt is $20$–$30 \times 10^6$ and $30 \times 10^6$ in weathered basalt.

(2) Surface weathering process: After the late Tertiary, the Shazi area entered the stage of rapid uplift and denudation. The basalt hydrolysates stored in the karst depression were weathered, leached, and employed in pedogenesis under the surface conditions. The soluble substances were carried out, and insoluble elements such as iron and scandium were relatively enriched. The grade of scandium in mineralized laterite is about 0.0007 wt.%, more than twice as much as fresh basalt. Previous studies have shown that less than 50% of the original rock material can be removed during weathering, which implies that the enrichment of insoluble elements can be doubled (the data are available in Supplementary Materials)—similar to the enrichment of Sc in Shazi laterite. Therefore, surface weathering is the main mechanism of scandium enrichment.

## 6. Conclusions

According to the variation of iron isotope composition and whole-rock iron content of magnetite in fresh basalt, weathered basalt, and laterite, combined with the previous research results, the genesis of Qinglong sand–laterite-type Sc deposit is the product of weathering hydrolysis of Emeishan basalt, and its mineralization process can be divided into the deep hydrolysis and surface weathering processes. In particular, surface weathering is an important mechanism leading to the relative enrichment of scandium.

**Supplementary Materials:** The following are available online at https://www.mdpi.com/article/10.3390/min11070737/s1, Figure S1. Typical XRD pattern of the fresh basalt in Shazi Sc deposit. Figure S2. Typical XRD pattern of the weathered basalt in Shazi Sc deposit. Figure S3. Typical XRD pattern of the laterite in Shazi Sc deposit. Figure S4. Geochemical evolution of a typical Sc-rich lateritic profile from Shazi deposit. Table S1. the compositions of different sections of the laterite profile in the shazi deposit.

**Author Contributions:** Conceptualization, J.S. and X.L.; methodology, Y.L.; investigation, J.S., X.L. and Y.L; writing—original draft preparation, J.S.; writing—review and editing, X.L. and Y.L. All authors have read and agreed to the published version of the manuscript.

**Funding:** This study was supported by the Natural Science Foundation of China (Grant No. 41262005), the joint fund of the science and technology department of Guizhou Province (No. LH[2014]7358), and the Startup Project of High-level Talents of the Guizhou Institute of Technology (No. XJGC20140702 and XJGC20131204).

**Data Availability Statement:** Not applicable.

**Conflicts of Interest:** The authors declare no conflict of interest.

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
