# Peer review of "Iron Isotope Constraints on the Mineralization Process of Shazi Sc-Rich Laterite Deposit in Qinglong County, China"

_minerals, doi:10.3390/min11070737_

Round 1
Reviewer 1 Report
Review comments on the manuscript minerals-1221840 entitled "Iron Isotope Constrain on the Mineralization Process of Shazi laterite-type Scandium Deposit in Qilong County, China". Submitted in Minerals under the title "The genesis of Shazi large-size anatase deposit in Qinglong, Western Guizhou, SW China"
This manuscript presents the results of iron isotope studies of magnetite separated from fresh basalts, weathered basalt and laterite.
The overall impression is average, the English text needs improvement, and the introduction should be expanded. I have included the pdf file with corrections and comments.
Introduction:
Scandium is a rare and indeed very important metal, which industrial demand increases for the last years. It is included in the list of critical raw materials not only from China but from European Union as well (since 2017). I think the authors should include the reference to it: https://ec.europa.eu/growth/sectors/raw-materials/specific-interest/critical_en. Here they can find the reports.
The authors should expand the text about scandium applications and can also include the following recent reference:
Wang et al. 2021 https://doi.org/10.1016/j.oregeorev.2020.103906
The authors may consider also the papers of Qin et al. 2020, 2021 https://doi.org/10.1016/j.chemgeo.2020.119771
https://doi.org/10.1016/j.gca.2020.11.020
Steffenssen et al. 2020 https://doi.org/10.1016/j.oregeorev.2020.103729
Halkoaho et al. 2020 https://doi.org/10.1007/s00126-020-00952-2
Scandium recovery: Wang et al. 2011 doi:10.1016/j.hydromet.2011.03.001
Beard et al. 2003 doi:10.1016/S0009-2541(02)00390-X
Discussion
5.2. Enrichment and mineralization of scandium
In this part the authors should include the papers of Qin et al. 2020, 2021 (given above) and edit the text accordingly.
For other corrections and comments see the pdf file.

Author Response
Manuscript Number: 12218410
Manuscript Title: Iron Isotope Constrain on the Mineralization Process of Shazi laterite-type Scandium Deposit in Qilong County, China
The authors would like to thank reviewers for their constructive comments and suggestions to improve the quality of the paper. We have studied comments carefully and have made corrections which we hope meet with approval. Revised portion are marked in red in paper. The main corrections in the paper and the responses are as flowing.
- Responses to Reviewer 1
(1) Review comments on the manuscript minerals-1221840 entitled "Iron Isotope Constrain on the Mineralization Process of Shazi laterite-type Scandium Deposit in Qilong County, China". Submitted in Minerals under the title "The genesis of Shazi large-size anatase deposit in Qinglong, Western Guizhou, SW China". This manuscript presents the results of iron isotope studies of magnetite separated from fresh basalts, weathered basalt and laterite. The overall impression is average, the English text needs improvement, and the introduction should be expanded. I have included the pdf file with corrections and comments.
Response: Thank you very much for your patience. According to your comments, we have corrected all the mistakes.
(2) Introduction:
Scandium is a rare and indeed very important metal, which industrial demand increases for the last years. It is included in the list of critical raw materials not only from China but from European Union as well (since 2017). I think the authors should include the reference to it: https://ec.europa.eu/growth/sectors/raw-materials/specific-interest/critical_en. Here they can find the reports.
The authors should expand the text about scandium applications and can also include the following recent reference:
Wang et al. 2021 https://doi.org/10.1016/j.oregeorev.2020.103906
The authors may consider also the papers of Qin et al. 2020, 2021 https://doi.org/10.1016/j.chemgeo.2020.119771
https://doi.org/10.1016/j.gca.2020.11.020
Steffenssen et al. 2020 https://doi.org/10.1016/j.oregeorev.2020.103729
Halkoaho et al. 2020 https://doi.org/10.1007/s00126-020-00952-2
Scandium recovery: Wang et al. 2011 doi:10.1016/j.hydromet.2011.03.001
Beard et al. 2003 doi:10.1016/S0009-2541(02)00390-X
Response: we have benefited a lot from the paper you provided. According to this suggestion, we have expanded the introduction appropriately.
Based on the enrichment processes, the primary Sc deposits can be divided into magmatic ore deposit, hydrothermal ore deposit, and supergene ore deposit [3]. The grade of Sc in these primary deposits is in the range of 0.005~0.04 wt.% [3]. Only 12 terrestrial minerals are known to contain Sc as an essential component, and thortveitite [Sc2Si2O7] is the most important among these minerals [4-5]. Significant proportions of Sc can be hosted in ferromagnesian silicate minereals (including clinopyroxene, garnet) and in a number of high field strength element (HFSE) minerals, such as wolframe, baddeleyite, rutile, etc [3,4,6-8]. At the beginning of the 20th century, Sc was mined from the thortveitite bearing pegmatite in Evje-Iveland, Norway [4]. Later, Sc was mainly recovered as by-product from residues, tailings ans waster liquors in the production of rare earth element (REE), tungsten and titanium [9]. Recently, about 90% of the global Sc production is from the Bayan Obo REE mine, the largest REE deposit in the world [4].
Rencently, The Sc-rich laterite were discovered in Cuba and Dominican Republic [10], Australia [11], New Caledonia [12], NE Argentina [13], and China [14], which is a potential Sc resource in the future. The laterite-type Sc deposit has attracted wide attention. On the one hand, the resource of this type of deposit is huge, and scandium can be mined as an independent deposit, on the other hand, the technology of extracting scandium from laterite is feasible and economical [11].
(3) Discussion
5.2. Enrichment and mineralization of scandium
In this part the authors should include the papers of Qin et al. 2020, 2021 (given above) and edit the text accordingly.
Response: we have read the above papers and revised our manuscript appropriately.
(4) For other corrections and comments see the pdf file.
Response: we have corrected all the mistakes.

Reviewer 2 Report
Dear authors
The subject of the research in the reviewed manuscript was magnetites separated from primary basalts and from the products of their weathering. The Fe isotope ratios were determined in the magnetite separates only and on this basis the authors describe the Sc concentration processes in the Shazi deposit. The planned and performed studies are not related to the discussed processes of Sc concentration in the studied deposit. Much of the discussion is based on literature data only. The authors of the manuscript do not present mineralogical or geochemical data documenting the mineral composition of the studied regoliths. In the discussion, they indicate the adsorption process of Sc on Fe oxyhydroxides, while in the paper they do not present elementary geochemical data (e.g. Sc concentrations in goethite-rich laterites or Sc content in separated goethite/hematite). Considerations in the discussion are not based on your own analytical data. The authors classify Shazi among the Sc deposits, although they previously described it as an anatase deposit. My task is that the Sc content in the laterites is too low, only 2 times higher than the content in the continental crust, which, in my opinion, does not qualify this deposit to the group of Sc deposits. Hence the calculation of the resource and the grade of the Sc ores makes no sense. At this point, I have a question on the basis of what data included in the paper, the authors calculated the Sc resources in the deposits, since I did not find a table with the results of chemical data in the paper!
The only analytical data presented in the paper are the ratios of Fe isotopes in the magnetite present in the studied weathering section. In my opinion, the presented data only authorize the discussion of Fe isotope fractionation during the Ti-magnetite leucoxenization processes. However, this task also requires careful mineralogical and petrographic research.
I conclude that the discussion in the paper is not based on a sufficient analytical basis, hence my decision is not to recommend the paper for publication in Minerals. Below I attach a few comments that will help the authors to prepare a new version of the manuscript.
line17-19: % of what? wt? mol?, at?
lines 32-33: Scandium is common in all rock types, of course in small amounts. Instead of indicating that increased concentrations are recorded in certain deposits, it is better to present the average contents in rocks, especially in those types which are then the subject of the authors' research.
line 39: why in capital letters? "Extracting Scandium"
line 42: "scandium exists in iron oxide in the form of adsorption" - unclear sentence
line 43: "iron isotope composition" - I think that analyzed are ratios only not composition!
lines 55-56: unclear sentence
line 74: sandstone is "lithic" by definition
lines 76-78: the sentences need to be redrafted
lines 81-82: chaotic description of the Maokou formation
lines 83-86: It is the first time that information about the Sc deposits appears, while the manuscript lacks chemical data allowing for the verification of these statements. There is no literature citation in the entire chapter on the geological setting of the study area. Does this mean that the authors of the manuscript are the only geologists working in the study area, they discovered the Sc deposits, they are the creators of the stratigraphic units/formations, they thoroughly performed all the petrographic studies themselves, etc.?
lines 87-89: There are no geochemical data in the paper, neither are the papers or other materials with these data cited. On what basis the authors have calculated the Sc resources in these deposits. In their earlier works, where the authors include chemical data, the authors describe the Shazi only as an anatase deposit.
lines 90, 95, 99: ore grade in scientific papers is shown as a percentage
line 99: "83.135 ×" ?
line 104: please add GPS coordinates of borehole no 1
lines 105-106: repeated information
line 141: It does not appear from the chapter on the research method that the limestones were sampled. Please supplement the chapter with the lithologic description of the examined samples and the depth of their collection in borehole No.1.
lines 179-189: The authors only determine the Fe isotopic ratios in magnetite. In the discussion, they describe complex transformations of minerals without petrographic and mineralogical control of magnetite and other components of weathered basalt. In what processes is secondary magnetite formed? What is the amount of this magnetite in relation to goethite/hematite in the studied samples? Please present the mineralogical documentation of this phase in the form of BSE and SEM images.
line 105: "hydrothermal"...?
line 202: In order to use an isotope geothermometer, we must have isotope ratios in two minerals between which Fe isotopes are fractionated (δ56Fe in pyroxene). The authors only determined the isotope ratios in the magnetite. The assumptions made led to unrealistically high temperatures (see line 204).
line 205: "magnetite and secondary magnetite" - the presence of secondary magnetite has not been mineralogically documented
line 210: "decomposition of clinopyroxene into secondary magnetite" - this is an unrealistic process under weathering conditions. Why is the secondary magnetite formed from clinopyroxene and not goethite or hematite, as indicated by the color of the sediments in the weathering section? Secondary goethite may also be responsible for the fractionation of Fe isotopes.
Lines 214-232: The discussion is pure speculation detached from the facts. There are no basic mineralogical data on the studied samples. The iron participating in the fractionation processes in the weathering environment can theoretically come from Fe-Ti oxides (magnetite, ilmenite), clinopyroxene, or olivine present in basalt. As a result of weathering alterations, Fe minerals from the smectite group, Fe oxyhydroxides, and/or hematite are formed. In this mineral system, the authors should discuss Fe isotope fractionation. There is no mineralogical documentation of secondary magnetite.
line 233: The discussion on Sc concentration is entirely a collage of literature data. The authors did not perform any mineralogical or chemical examinations of the minerals present in the studied weathering section. The Sc concentrations in the tested samples are also unknown. The conclusion that Sc is adsorbed on Fe oxyhydroxides is unsupported by analytical data, even the simplest ones, allowing to assess the difference in Sc concentration in the zones rich and depleted in goethite.
Author Response
Responses to Reviewer’s Comments on our paper
Manuscript Number: 12218410
Manuscript Title: Iron Isotope Constrain on the Mineralization Process of Shazi laterite-type Scandium Deposit in Qilong County, China
The authors would like to thank reviewers for their constructive comments and suggestions to improve the quality of the paper. We have studied comments carefully and have made corrections which we hope meet with approval. Revised portion are marked in red in paper. The main corrections in the paper and the responses are as flowing.
- Responses to Reviewer 2
The subject of the research in the reviewed manuscript was magnetites separated from primary basalts and from the products of their weathering. The Fe isotope ratios were determined in the magnetite separates only and on this basis the authors describe the Sc concentration processes in the Shazi deposit. The planned and performed studies are not related to the discussed processes of Sc concentration in the studied deposit. Much of the discussion is based on literature data only. The authors of the manuscript do not present mineralogical or geochemical data documenting the mineral composition of the studied regoliths. In the discussion, they indicate the adsorption process of Sc on Fe oxyhydroxides, while in the paper they do not present elementary geochemical data (e.g. Sc concentrations in goethite-rich laterites or Sc content in separated goethite/hematite). Considerations in the discussion are not based on your own analytical data. The authors classify Shazi among the Sc deposits, although they previously described it as an anatase deposit. My task is that the Sc content in the laterites is too low, only 2 times higher than the content in the continental crust, which, in my opinion, does not qualify this deposit to the group of Sc deposits. Hence the calculation of the resource and the grade of the Sc ores makes no sense. At this point, I have a question on the basis of what data included in the paper, the authors calculated the Sc resources in the deposits, since I did not find a table with the results of chemical data in the paper!
The only analytical data presented in the paper are the ratios of Fe isotopes in the magnetite present in the studied weathering section. In my opinion, the presented data only authorize the discussion of Fe isotope fractionation during the Ti-magnetite leucoxenization processes. However, this task also requires careful mineralogical and petrographic research.
I conclude that the discussion in the paper is not based on a sufficient analytical basis, hence my decision is not to recommend the paper for publication in Minerals. Below I attach a few comments that will help the authors to prepare a new version of the manuscript.
Response: In the revised version, we provide a typical schematic diagram of laterite profile and the distribution of elements in the profile. The mineralogical compositions of basalt, weathered basalt and laterite were determined by XRD.
In generally, magmatic rocks with average Sc concentrations >60 ppm can be regarded as Sc deposits. The Sc cut-off grades of laterites vary from 33 ppm in the Shazi deposit, China (Nie et al., 2018), 70 ppm in the Lucknow deposit, Australia, to 300 ppm in the Syerston-Flemington deposit, Australia.
Reference:
Nie, A., Zhang, M. and Sun, J. (2018) The first discovery of a large independent scandium deposit formed by sparse element scandium in guizhou. Journal of Guizhou University (Natural Sciences) 35, 8-13 (in Chinese with English abstract).
(2) line17-19: % of what? wt? mol?, at?
Response: the unit should be wt.%.
(3) lines 32-33: Scandium is common in all rock types, of course in small amounts. Instead of indicating that increased concentrations are recorded in certain deposits, it is better to present the average contents in rocks, especially in those types which are then the subject of the authors' research.
Response: we expand the introduction.
(4) line 39: why in capital letters? "Extracting Scandium"
Response: we have modified these mistakes.
(5) line 42: "scandium exists in iron oxide in the form of adsorption" - unclear sentence
Response: we have written this sentence.
scandium is adsorbed on goethite in the laterite
(6) line 43: "iron isotope composition" - I think that analyzed are ratios only not composition!
Response: we have deleted “composition”
(7) lines 55-56: unclear sentence
Response: we have rewritten this sentence
Based on the iron isotope study of magnetite in fresh basalt, weathered basalt and laterite in the area, combined with the whole rock iron content data, this paper aims to find out the changes of iron isotope and iron content in basalt lateritization process, and to investigate the enrichment process of scandium.
(8) line 74: sandstone is "lithic" by definition
Response: we deleted “lithic”
(9) lines 76-78: the sentences need to be redrafted
Response: we have redrafted this sentence.
It has pseudoconformity contact with the underlying Upper Permian Emeishan basalt formation. It is composed of basaltic lava, basaltic lava breccia, pyroclastic rock, tholeiitic basalt and tuff. Emeishan basalt is widely distributed in the study area.
(10) lines 81-82: chaotic description of the Maokou formation
Response: we have rewritten this sentence.
The Middle Permian Maokou Formation is distributed in most areas of the study area. The lithology is bright crystal bioclastic limestone and limestone, and these limestones are gray and dark gray in color.
(11) lines 83-86: It is the first time that information about the Sc deposits appears, while the manuscript lacks chemical data allowing for the verification of these statements. There is no literature citation in the entire chapter on the geological setting of the study area. Does this mean that the authors of the manuscript are the only geologists working in the study area, they discovered the Sc deposits, they are the creators of the stratigraphic units/formations, they thoroughly performed all the petrographic studies themselves, etc.?
Response: The Shazi Sc deposit was discovered by Ni Aiguo, the doctoral supervisor of the first author. The research object of the doctoral dissertation of the first author is the Shazi Sc deposit. Unfortunately, no relevant contributions in English have been published. In this paper, we cite Ni et al. (2018) article as a reference.
(12) lines 87-89: There are no geochemical data in the paper, neither are the papers or other materials with these data cited. On what basis the authors have calculated the Sc resources in these deposits. In their earlier works, where the authors include chemical data, the authors describe the Shazi only as an anatase deposit.
Response: Response: The contents of TiO2 and Sc2O3 in the Shazi laterite-type deposit are both very high, and the resources of titanium and scandium have reached the level of large-scale deposits. However, in this deposit, titanium mainly exists in the form of anatase, which is difficult to utilize at the current level. In this paper, we call it Shazi laterite-type scandium deposit.
In this revised version, we cited Ni et al. (2018)
(13) lines 90, 95, 99: ore grade in scientific papers is shown as a percentage
Response: we follow this suggestion.
(14) line 99: "83.135 ×" ?
Response: it should be 0.0083 wt.%
(15) line 104: please add GPS coordinates of borehole no 1
Response: we have added GPS coordinates of Borehole No.1
(16) lines 105-106: repeated information
Response: we have rewritten these sentences
Eight Samples were collected from drill hole in No.1 ore body (N25°46′40.72″; E105°9′0.01″): two samples from fresh pillow basalt, two samples weathed basalt and four from laterite.
(17) line 141: It does not appear from the chapter on the research method that the limestones were sampled. Please supplement the chapter with the lithologic description of the examined samples and the depth of their collection in borehole No.1.
Response: The Fe isotope of the limestone is not included in this paper.
(18) lines 179-189: The authors only determine the Fe isotopic ratios in magnetite. In the discussion, they describe complex transformations of minerals without petrographic and mineralogical control of magnetite and other components of weathered basalt. In what processes is secondary magnetite formed? What is the amount of this magnetite in relation to goethite/hematite in the studied samples? Please present the mineralogical documentation of this phase in the form of BSE and SEM images.
Response: We are very sorry that there is no mineralogical evidence of the sample in this paper. In the revised version, we provided the XRD of the samples (see appendix 1). According to XRD of the sample, magnetite is the only iron oxide in the weathered basalt.
(19) line 105: "hydrothermal"...?
Response: the word “fluid” is missing. The phrase should be “hydrothermal fluid”
(20) line 202: In order to use an isotope geothermometer, we must have isotope ratios in two minerals between which Fe isotopes are fractionated (δ56Fe in pyroxene). The authors only determined the isotope ratios in the magnetite. The assumptions made led to unrealistically high temperatures (see line 204).
Response: Chen et al. (2014) analyzed the Fe isotopic composition of clinopyroxene from the Baima gabbro layered intrusions, which belong to the Emeishan basalt Formation. In this work, we cited their data.
After the eruption of Emeishan basalt, Shazi area experienced a stage of rapid burial and then slow denudation, with a burial history of more than 100 Ma. The burial depth of Qinglong Antimony Deposit near Shazi is about 2 km, and the burial temperature is about 100 ℃, so we think that the temperature calculated from the iron isotope between pyroxene and magnetite is reasonable
(21) line 205: "magnetite and secondary magnetite" - the presence of secondary magnetite has not been mineralogically documented
Response: We really have no direct evidence for the formation of secondary magnetite. We infer the existence of secondary magnetite based on the Fe isotope of magnetite om weathered basalt and laterite.
(22) line 210: "decomposition of clinopyroxene into secondary magnetite" - this is an unrealistic process under weathering conditions. Why is the secondary magnetite formed from clinopyroxene and not goethite or hematite, as indicated by the color of the sediments in the weathering section? Secondary goethite may also be responsible for the fractionation of Fe isotopes.
Response: As mentioned above, magnetite is the only iron oxide in weathered basalt.
(23) Lines 214-232: The discussion is pure speculation detached from the facts. There are no basic mineralogical data on the studied samples. The iron participating in the fractionation processes in the weathering environment can theoretically come from Fe-Ti oxides (magnetite, ilmenite), clinopyroxene, or olivine present in basalt. As a result of weathering alterations, Fe minerals from the smectite group, Fe oxyhydroxides, and/or hematite are formed. In this mineral system, the authors should discuss Fe isotope fractionation. There is no mineralogical documentation of secondary magnetite.
Response: According to XRD (Appendix), fresh basalt is mainly composed of clinopyroxene and plagioclase, and iron oxides include magnetite. In addition to clinopyroxene, plagioclase and magnetite, montmorillonite appears in weathered basalt, while laterite is mainly composed of clay minerals and iron (oxyhydr)oxides (such as geothite and hematite). Magnetite appears in heavy sand minerals. Based on the existing evidence, magnetite in basalt is obviously different from the other Fe-bearing minerals in isotopic composition, magnetite is the only mineral existing in all three lithologies, and the isotope of magnetite has changed significantly. Therefore,the variation of iron isotope in magnetite is an effective indicator of whole rock iron isotope
(24) line 233: The discussion on Sc concentration is entirely a collage of literature data. The authors did not perform any mineralogical or chemical examinations of the minerals present in the studied weathering section. The Sc concentrations in the tested samples are also unknown. The conclusion that Sc is adsorbed on Fe oxyhydroxides is unsupported by analytical data, even the simplest ones, allowing to assess the difference in Sc concentration in the zones rich and depleted in goethite.
Response: It has been proved that scandium is mainly adsorbed in goethite (Chassé et al., 2017; 2019; Qin et al., 2020; 2021).
In the revised version, we provide a typical schematic diagram of laterite profile and the distribution of elements in the profile
References:
(1) Chassé, M., Griffin, W.L., O'Reilly, S.Y. and Calas, G. (2017) Scandium speciation in a world-class lateritic deposit. Geochemical Perspective Letters 3, 105-114.
(2) Chassé, M., Griffin, W.L., O’Reilly, S.Y. and Calas, G. (2019) Australian laterites reveal mechanisms governing scandium dynamics in the critical zone. Geochimica et Cosmochimica Acta 260, 292-310.
(3) Qin, H.-B., Yang, S., Tanaka, M., Sanematsu, K., Arcilla, C. and Takahashi, Y. (2020) Chemical speciation of scandium and yttrium in laterites: New insights into the control of their partitioning behaviors. Chemical Geology 552, 119771.
(4) Qin, H.-B., Yang, S., Tanaka, M., Sanematsu, K., Arcilla, C. and Takahashi, Y. (2021) Scandium immobilization by goethite: Surface adsorption versus structural incorporation. Geochimica et Cosmochimica Acta 294, 255-272.

Reviewer 3 Report
Is any relationship between the "Shazi laterite-type scandium deposit" and "Shazi large-sized anatase deposit"? Are they the same deposit? Not all the readers are familiar with the geology of China and its mineral deposits. Please, elaborate.
Line 49: "a large Ti-Sc deposit" The same problem as above. Please, elaborate the relationship.
Fig. 1: "attitude" has another meaning. It is better to use "dip". Also, the pattern of the orebody shown on the map is different from the one from the legend.
Subchapter 3.2 Analytical Methods: There is no citation. I think, the authors learn about the method from another article or book. Correct is to cite the work were the methods has been described in details.
Line 126: HR-MC-ICPMS. When you mention a method for the first time in the paper, you do not use the abbreviation. So, write High Resolution-Multiple Collector-... and the abbreviation between brackets.
Line 138: for the readers that are not familiar with the determination of iron isotopes, I suggest to detail what IRMM... stands for. Much better is to elaborate a bit the paragraph from line 137 to line 139.
Lines 142, 143, 146: 56 before Fe has to be superscript.
Figure 3: Caption: T stands for total Fe.
Line 241: Chasse et al... The "c" has to be capital "C".
Author Response
Responses to Reviewer’s Comments on our paper
Manuscript Number: 12218410
Manuscript Title: Iron Isotope Constrain on the Mineralization Process of Shazi laterite-type Scandium Deposit in Qilong County, China
The authors would like to thank reviewers for their constructive comments and suggestions to improve the quality of the paper. We have studied comments carefully and have made corrections which we hope meet with approval. Revised portion are marked in red in paper. The main corrections in the paper and the responses are as flowing.
- Responses to Reviewer 3
(1) Is any relationship between the "Shazi laterite-type scandium deposit" and "Shazi large-sized anatase deposit"? Are they the same deposit? Not all the readers are familiar with the geology of China and its mineral deposits. Please, elaborate.
Line 49: "a large Ti-Sc deposit" The same problem as above. Please, elaborate the relationship.
Response: The contents of TiO2 and Sc2O3 in the Shazi laterite-type deposit are both very high, and the resources of titanium and scandium have reached the level of large-scale deposits. However, in this deposit, titanium mainly exists in the form of anatase, which is difficult to utilize at the current level. In this paper, we call it Shazi laterite-type scandium deposit
(3) Fig. 1: "attitude" has another meaning. It is better to use "dip". Also, the pattern of the orebody shown on the map is different from the one from the legend.
Response: we have change “attitude” to “dip”. In this paper, “ore body” is used.
(4) Subchapter 3.2 Analytical Methods: There is no citation. I think, the authors learn about the method from another article or book. Correct is to cite the work were the methods has been described in details.
Response: we follow this suggestion.
Fe isotope analysis were done at the State Key Laboratory of Ore deposit Geochemistry, Institute of Geochemistry, Chinese Academy of Sciences. The detailed procedures for sample dissolution, column chemistry and instrumental analysis were introducted in detail by Zhu et al. (2008). Only a brief description is given below.
(5) Line 126: HR-MC-ICPMS. When you mention a method for the first time in the paper, you do not use the abbreviation. So, write High Resolution-Multiple Collector-... and the abbreviation between brackets.
Response: we follow this suggestion.
(6) Line 138: for the readers that are not familiar with the determination of iron isotopes, I suggest to detail what IRMM... stands for. Much better is to elaborate a bit the paragraph from line 137 to line 139.
Response: we follow this suggestion.
For iron isotope, IRMM-014 provided by the Institute of reference materials and measurement of the European Commission is often used as the isotope standard. The Fe isotopic abundance of IRMM-014 is as follows: 5.845±0.023% (54Fe)、91.754±0.024% (56Fe)、2.1192±0.0065% (57Fe)、0.2818±0.0027% (58Fe) (Taylor et al., 1992).
(7) Lines 142, 143, 146: 56 before Fe has to be superscript.
Response: we have corrected these errors.
(8) Figure 3: Caption: T stands for total Fe.
Response: we follow this suggestion.
(9) Line 241: Chasse et al... The "c" has to be capital "C".
Response: we have corrected this error.
Reviewer 4 Report
The authors have presented a very short paper on the Fe isotope variability in the Shazi Sc deposit, Qilong, China.
Although, traditionally, laterite deposits can be very complex mineralogically, I would have liked the authors to include a bit more about deportment of certain elements, particularly Sc given it is a Sc deposit.
The authors use the difference in Fe isotopes to conclude that hydrolysis of pyroxene in the basalt is responsible. They do mention an increase in Sc in the laterite relative to basalt due to surficial weathering, assuming that pyroxene and hornblende (and potentially Fe-Ti minerals such as titanite?) are the main hosts. However, there is a lack of information regarding mineralogy of both the basalt (source in this case), changes in the weathered basalt and then laterite. This might provide additional evidence to support their results and discussion.
It would also be beneficial to provide the readers with some images, either of the laterite deposits themselves (e.g. with the various laterite horizons labelled) and/or microscopy images (optical or SEM).
The paper also needs some English phrasing and editing changes.
Together with these additions, I feel the paper would benefit and the authors' conclusions would be greater supported. As it stands, the paper needs to be a bit more holistic in its approach. I will leave it with the editor to decide whether these additions can be made within an appropriate timeframe.
Author Response
Responses to Reviewer’s Comments on our paper
Manuscript Number: 12218410
Manuscript Title: Iron Isotope Constrain on the Mineralization Process of Shazi laterite-type Scandium Deposit in Qilong County, China
The authors would like to thank reviewers for their constructive comments and suggestions to improve the quality of the paper. We have studied comments carefully and have made corrections which we hope meet with approval. Revised portion are marked in red in paper. The main corrections in the paper and the responses are as flowing.
- Responses to Reviewer 4
(1) The authors have presented a very short paper on the Fe isotope variability in the Shazi Sc deposit, Qilong, China. Although, traditionally, laterite deposits can be very complex mineralogically, I would have liked the authors to include a bit more about deportment of certain elements, particularly Sc given it is a Sc deposit.
The authors use the difference in Fe isotopes to conclude that hydrolysis of pyroxene in the basalt is responsible. They do mention an increase in Sc in the laterite relative to basalt due to surficial weathering, assuming that pyroxene and hornblende (and potentially Fe-Ti minerals such as titanite?) are the main hosts. However, there is a lack of information regarding mineralogy of both the basalt (source in this case), changes in the weathered basalt and then laterite. This might provide additional evidence to support their results and discussion.
It would also be beneficial to provide the readers with some images, either of the laterite deposits themselves (e.g., with the various laterite horizons labelled) and/or microscopy images (optical or SEM).
The paper also needs some English phrasing and editing changes.
Together with these additions, I feel the paper would benefit and the authors' conclusions would be greater supported. As it stands, the paper needs to be a bit more holistic in its approach. I will leave it with the editor to decide whether these additions can be made within an appropriate timeframe.
Response: In the revised version, we provide a typical schematic diagram of laterite profile and the distribution of elements in the profile. The mineralogical compositions of basalt, weathered basalt and laterite were determined by XRD.

Round 2
Reviewer 2 Report
Dear authors,
Thank you for sending the revised version of the manuscript. All the minor remarks were accepted by the authors and conscientiously introduced into the manuscript. Unfortunately, the changes are of a cosmetic nature and do not remove the main drawback of the paper, i.e. the lack of relationship between the analytical data and the discussion and conclusions presented in the article. In my opinion, the time that has elapsed since my answer is not sufficient to perform the necessary geochemical and mineralogical studies. The only analytical data is the information (from the XRD examination) about the composition of the Emeishan basalt. The conducted research was not recorded in the chapter "material and research methods", moreover, the given composition is typical for basalt rocks all over the world. The authors did not refer to the presence of ilmenite in the basalt, which is a very common component of such rocks, but identifies enigmatically Fe-Ti oxides, among which magnetite dominates.
I am not convinced by the Sc resources calculated in the dicovered deposits, the analytical data quoted by the authors are published in a local scientific journal. Most geologists do not have access to these data. I suggest the authors first publish a complete set of analytical data and describe the newly discovered Sc deposits in one of the mainstream magazines
As a strange situation, I also perceive drawing a conclusion about the presence of secondary magnetite on the basis of the results of isotope studies. The best way to document the presence of secondary magnetite is to perform detailed petrographic and mineralogical studies. The texture of magnetite is the best source of data allowing to distinguish different generations of this mineral.
The authors study magnetites only that were separated from the basalts and their weathering cover. There is no diffraction control of the purity of magnetite selected for isotope examination in the study. How ilmenite was identified, the presence of which is very likely in the analyzed magnetite samples selected for the research. How was ilmenite separated from magnetite? What is the influence of weathering processes on the leucoxenization of the studied Ti-rich magnetite and ilmenite? The lack of conscientiously prepared petrographic documentation (reflected light microscopy, electron microscopy - BSE images, EPMA) has a very strong impact on my negative assessment of the work submitted for review.
In connection with the arguments presented above, it maintains my original opinion.
With the best regards
Author Response
Responses to Reviewer’s Comments on our paper
Manuscript Number: 12218410
Manuscript Title: Iron Isotope Constrain on the Mineralization Process of Shazi laterite-type Scandium Deposit in Qilong County, China
The authors would like to thank reviewers for their constructive comments and suggestions to improve the quality of the paper. Revised portion are marked in red in paper.
Thank you for sending the revised version of the manuscript. All the minor remarks were accepted by the authors and conscientiously introduced into the manuscript. Unfortunately, the changes are of a cosmetic nature and do not remove the main drawback of the paper, i.e., the lack of relationship between the analytical data and the discussion and conclusions presented in the article. In my opinion, the time that has elapsed since my answer is not sufficient to perform the necessary geochemical and mineralogical studies. The only analytical data is the information (from the XRD examination) about the composition of the Emeishan basalt. The conducted research was not recorded in the chapter "material and research methods", moreover, the given composition is typical for basalt rocks all over the world. The authors did not refer to the presence of ilmenite in the basalt, which is a very common component of such rocks, but identifies enigmatically Fe-Ti oxides, among which magnetite dominates.
I am not convinced by the Sc resources calculated in the discovered deposits, the analytical data quoted by the authors are published in a local scientific journal. Most geologists do not have access to these data. I suggest the authors first publish a complete set of analytical data and describe the newly discovered Sc deposits in one of the mainstream magazines
As a strange situation, I also perceive drawing a conclusion about the presence of secondary magnetite on the basis of the results of isotope studies. The best way to document the presence of secondary magnetite is to perform detailed petrographic and mineralogical studies. The texture of magnetite is the best source of data allowing to distinguish different generations of this mineral.
The authors study magnetites only that were separated from the basalts and their weathering cover. There is no diffraction control of the purity of magnetite selected for isotope examination in the study. How ilmenite was identified, the presence of which is very likely in the analyzed magnetite samples selected for the research. How was ilmenite separated from magnetite? What is the influence of weathering processes on the leucoxenization of the studied Ti-rich magnetite and ilmenite? The lack of conscientiously prepared petrographic documentation (reflected light microscopy, electron microscopy - BSE images, EPMA) has a very strong impact on my negative assessment of the work submitted for review.
Response: the suggestions and comments of the reviewer have great implication for our future research work. We will carry out future research on the mineralogy of laterite.

Reviewer 4 Report
The authors need to have the English checked/edited
Author Response
Responses to Reviewer’s Comments on our paper
Manuscript Number: 12218410
Manuscript Title: Iron Isotope Constrain on the Mineralization Process of Shazi laterite-type Scandium Deposit in Qilong County, China
The authors would like to thank reviewers for their constructive comments and suggestions to improve the quality of the paper. Revised portion are marked in red in paper.
(1) The authors need to have the English checked/edited
Response: This version has undergone English language editing by MDPI.

This manuscript is a resubmission of an earlier submission. The following is a list of the peer review reports and author responses from that submission.